# The Role of *Substance P* in Corneal Homeostasis

**DOI:** 10.3390/biom15050729

**Published:** 2025-05-16

**Authors:** Jastrzębska-Miazga Irmina, Machna Bartosz, Wyględowska-Promieńska Dorota, Smędowski Adrian

**Affiliations:** 1Department of Ophthalmology, Kornel Gibinski University Clinical Center, Medical University of Silesia, 40-514 Katowice, Poland; irmina.jastrzebska@gmail.com (J.-M.I.);; 2Department of Ophthalmology, Faculty of Medical Sciences in Katowice, Medical University of Silesia, 40-055 Katowice, Poland; bartoszmachna@gmail.com; 3Laboratory for Translational Research in Ophthalmology, Department of Ophthalmology, Kornel Gibinski University Clinical Center, Medical University of Silesia, 40-055 Katowice, Poland; 4Department of Pediatric Ophthalmology, Faculty of Medical Sciences in Katowice, Medical University of Silesia, 40-055 Katowice, Poland; 5Department of Pediatric Ophthalmology, Kornel Gibinski University Clinical Center, Medical University of Silesia, 40-055 Katowice, Poland; 6GlaucoTech Co., 40-282 Katowice, Poland

**Keywords:** substance P, corneal homeostasis, neurokinin-1-receptor, corneal innervation

## Abstract

The cornea, a highly innervated and avascular ocular tissue, relies on intricate neuro-immune interactions to maintain homeostasis. Among key neuromediators, substance P (SP)—a neuropeptide belonging to the tachykinin family—plays a dual role in corneal physiology and pathology. This review synthesizes current knowledge on SP’s involvement in corneal innervation, epithelial homeostasis, immune regulation, neovascularization, and wound healing, while highlighting its dichotomous effects in both promoting tissue repair and exacerbating inflammation. SP, primarily signaling through the neurokinin-1 receptor (NK1R), influences corneal epithelial proliferation, barrier function, and wound healing by modulating cytokines, chemokines, and growth factors. However, its overexpression is linked to pain sensitization, inflammatory keratitis, and corneal neovascularization, driven by interactions with immune cells (e.g., mast cells, neutrophils) and pro-angiogenic factors (e.g., VEGF). Clinical studies demonstrate altered SP levels in dry eye disease, neurotrophic keratitis, and post-refractive surgery, correlating with nerve damage and ocular surface dysfunction. Emerging therapies targeting SP pathways- such as NK1R antagonists (e.g., fosaprepitant) and SP-IGF-1 combinations-show promise for treating neurotrophic ulcers but face challenges due to SP’s context-dependent actions. Future research should clarify the roles of NK2R/NK3R receptors and optimize SP-based interventions to balance its reparative and inflammatory effects. Understanding SP’s multifaceted mechanisms could advance the development of therapies for corneal diseases, particularly those involving sensory neuropathy and immune dysregulation.

## 1. Introduction

The cornea, the anterior part of the eye, is an avascular layer composed of epithelial cells, endothelial cells, fibroblasts, collagen fibrils, nerve fibers, and immune cells. It provides refractive power, structural integrity, and defense against microbes and environmental factors. It has the richest innervation of all body tissues, primarily supplied by the nasociliary nerve of the ophthalmic branch (V1) of the trigeminal nerve. Occasionally, sensory nerves from the maxillary branch (V2) innervate the inferior cornea [1]. Additionally, autonomic innervation from the superior cervical ganglion (sympathetic) and ciliary ganglion (parasympathetic) is present, although the role and density of these nerves remain unclear [2]. It is estimated that the subbasal plexus of the cornea contains approximately 19,000–44,000 axons [2]. Corneal innervation is critical for delivering trophic factors (e.g., substance P, calcitonin gene-related peptide, acetylcholine, serotonin, neuropeptide Y) which help to maintain health of the corneal epithelium. Corneal homeostasis relies on the relationship between corneal nerves and epithelium—epithelial cells secrete neurotrophic factors (e.g., nerve growth factor and ciliary neurotrophic factor) that influence corneal nerves [2]. Numerous studies have shown that loss of corneal sensory innervation leads to a decrease of the vitality, metabolism, and reduction of the mitotic activity of epithelial cells, resulting in epithelial breakdown [3,4]. The thickness of corneal epithelium decreases, and epithelial cells exhibit intracellular swelling, loss of microvilli, and an abnormal production of the basal lamina. This can lead to many pathologies such as recurrent or persistent epithelial defects, ulceration, stromal melting, and subsequent perforation [5]. Several studies have focused on the role of sensory neuromediators in the corneal epithelium pathophysiology. These studies have shown, among other findings, the depletion of the substance P (SP) and acetylcholine (Ach) in the rat cornea after sensory nerve injury [6].

In vitro, SP, cholecystokinin gene-related peptide (CGRP), and Ach maintain proper epithelial proliferation. According to experimental and clinical evidence, there is a bidirectional control of corneal epithelium proliferation: sensory neuromediators promote epithelial cell mitosis, while sympathetic mediators, such as epinephrine and norepinephrine, reduce epithelial cell mitosis. Cavanagh HD et al. proposed that endogenous proliferation of corneal epithelial cells is regulated by the process characterized by an adrenergic, cAMP-dependent off-, and a cholinergic, muscarinic cGMP-dependent on-response. Exogenous substances that increase intracellular cAMP levels, such as isoproterenol or PGE1, inhibit epithelial mitosis. Conversely, carbamylcholine or ACH raise intranuclear cGMP levels and increase mitosis by regulatory stimulation of RNA-polymerase II activity [7].

Substance P (SP) is a peptide present in both, the central and the peripheral nervous system. It is an undecapeptide, member of tachykinin family, discovered by Von Euler and Gaddum [8]. It is produced by various cells, including epithelial and endothelial cells, neurons, astrocytes, microglia, and immune cells such as eosinophils, dendritic cells (DCs), and T-cells [9,10,11]. SP exerts its biological and immunological activity through high-affinity neurokinin 1 receptor (NK1R), neurokinin 2 receptor (NK2R) and neurokinin 3 receptor (NK3R) [12]. It shows the highest affinity for NK1R, which is encoded by TAC1R on chromosome 2. There are two isoforms of NK1R receptor found in neuronal and immune cells—the full-length NK1R and the truncated NK1R (NK1R-T) [13]. NK1R, primarily found in the central nervous system, has a ten-fold higher affinity to substance P than its truncated isoform, which is predominantly present in peripheral tissues such as heart, lung, spleen, prostate and immune cells [14].

Substance P is encoded by the Tac1 (pre-protachkinin-A) gene, located on chromosome 7. It consists of an eleven-amino-acid sequence “H-Arg1-Pro2-Lys3-Pro4-Gln5-Gln6-Phe7-Phe8-Gly9-Leu10-Met11-NH2”. Due to its amphiphilic properties, attributed to a polar, positively charged N-terminal and non-polar, uncharged C-terminus, it can interact with the phospholipid bilayer of the plasma membrane [15]. Substance P is abundantly produced by numerous cell types throughout the body and, as a result, mediates diverse cell-specific physiological, as well as, pathological functions [16].

## 2. The Role of Substance P in Corneal Pain Perception

The release of substance P from nociceptive nerve fibers and activation of its receptor, neurokinin 1, are crucial in the transmission of pain signals. Its role in the regulation of nociceptive information at the first sensory synapse in the spinal cord is well-documented [17]. Elevated levels of substance P have been reported in various painful corneal pathologies, such as infectious keratitis or dry eye disease. Moreover, even the placement of an intrastromal suture has been shown to increase its levels [18]. Studies have demonstrated that mice lacking SP exhibit significantly reduced corneal pain sensation. Additionally, topical administration of the NK1R antagonist—fosaprepitant—in mice reduced ocular surface nociception by decreasing SP release in tears [19]. Another study demonstrated that topical fosaprepitant significantly reduced pain in patients suffering from chronic ocular pain and inflammation [20].

Pioneering work by Belmonte, Marfurt, and others has also elucidated the complex interplay between SP and corneal nerve function [21,22]. Notably, SP interacts with TRPV1 (the capsaicin receptor), a key mediator of thermal and chemical pain sensitivity [23,24,25]. TRPV1 activation enhances SP release, amplifying inflammatory and pain responses [26]. This mechanism is critical in conditions like dry eye disease and neuropathic corneal pain, where elevated SP levels correlate with hyperalgesia. Further research is needed to clarify how SP/TRPV1 crosstalk influences corneal sensory pathways.

On the other hand, several recent studies have indicated that SP also exhibits anti-nociceptive properties in the peripheral nervous system [27]. One study revealed an unexpected antinociceptive effect of substance P against chronic mechanical hyperalgesia, induced by repeated intramuscular acid injection [28].

## 3. The Role of Substance P in Regulating Immune Response

As Substance P (SP) is expressed by neurons, astrocytes and microglia, it is present in both the central and peripheral nervous system. Although it is considered a neuropeptide, SP is also produced by immune cells and acts in an autocrine or paracrine manner to regulate immune cell function. Nerves releasing SP are reported to innervate primary (thymus and bone-marrow) and secondary (spleen, lymph nodes, tonsils, and gut-associated lymphoid tissue) lymphoid organs [29].

SP and its receptor NK1R are also well-documented to be expressed in various immune cell types. Dendritic cells, monocytes/macrophages, eosinophils, neutrophils, natural killer cells and T lymphocytes in both humans and rodents have been shown to express SP [30,31,32,33,34]. The role of substance P in viral, bacterial, and parasitic infection has been also a subject of investigation [35,36].

SP modulates the immune response through a wide range of mechanisms. It upregulates expression of IL-2, which enhances lymphocyte proliferation and stimulates immunoglobulin production [37]. Upregulation of the Wnt signaling pathway—another effect of substance P—results in enhanced proliferation of bone marrow stromal cells [38].

Furthermore, substance P stimulates the production of chemokines such as CCL4, CCL5, CXCL2, MCP-1 and IL-8 which induces the recruitment of immune cells [39]. It also upregulates adhesion molecules—MAC-1 and its ligand ICAM-1—present on NK1R+ dendritic cells [40]. Those processes lead to further amplification of inflammatory responses.

Substance P enhances the proliferation and recruitment of many immune cells and mediates various functions. For example, it activates eosinophils, causing their degranulation and superoxide release [41]. Raap and colleagues demonstrated that substance P- induced Ca^2+^ influx prevents eosinophil apoptosis [31].

By increasing production of chemotactic cytokines, as mentioned earlier, substance P enhances neutrophil chemotaxis and induces phagocytic activity [42].

Notably, a preclinical study demonstrated the therapeutic potential of topical NK-1 receptor antagonist fosaprepitant in a murine model of ocular graft-versus-host disease (GVHD). This study highlights NK1R signaling as a promising therapeutic target for ocular GVHD, with fosaprepitant showing efficacy against both inflammatory and epithelial components of this vision-threatening condition [19].

Substance P is also known for its pro-angiogenic effects. It induces angiogenesis directly by promoting nitric oxide production in endothelial cells and indirectly through the induction of mast cell degranulation and secretion of VEGF (vascular endothelial growth factor) and TNF-alpha [43,44].

Several studies have demonstrated a surprising role of substance P and the neurokinin-1 receptor in the host response in viral infections [45]. Specifically, the presence of substance P can augment replication of HIV in cultured macrophages, which is particularly significant given that levels of this tachykinin are elevated in patients with this retroviral disease. Furthermore, rodent models of paramyxovirus infection have shown that the presence of neurokinin receptors and their ligands contributes to the destructive inflammatory response in airways [35].

SP has also been reported to regulate virus- and bacteria-induced inflammation in the cornea. In a mouse model of corneal herpes simplex virus-1 infection, SP was shown to regulate the severity of herpes stromal keratitis (HSK) induced by the herpes simplex virus-1 infection [46]. In fact, numerous reports have highlighted the role of SP in ocular inflammation [47]. For example, human samples from patients with pterygium showed altered expression of SP and NK1R in pterygium fibroblasts. Moreover, cell culture studies indicate that SP, via NK1R, induces the migration of pterygium fibroblasts and microvascular endothelial cells, suggesting that SP may contribute to the pathogenesis of pterygia through its profibrogenic and angiogenic action [48]. In patients with allergic conjunctivitis, higher levels of SP were found in the tear fluid [49]. Treatment of conjunctivitis in patients with ocular allergies and in animal models has been shown to reduce the SP level in tears [50].

## 4. The Role of Substance P in Corneal Neovascularization

Another area of interest is the role of SP in stimulating corneal neovascularization through NK1R [51]. Substance P can be detected in epithelial cells and nerves of the healthy cornea, but under normal conditions, the cornea remains avascular [2]. The problem arises when injured corneal tissues trigger the release of SP from nerves, leading to inflammation and neovascularization [52]. The underlying mechanism may involve the direct action of SP on vascular endothelial cells—promoting endothelial cell proliferation through the fibroblast growth factor-beta and nitric oxide pathways [44]. It may also indirectly promote hemangiogenesis by recruiting granulocytes with angiogenic potential from the bloodstream [53]. Furthermore, substance P has been shown to induce pro-angiogenic phenotype shift in most leukocyte populations. Through NK1R, it stimulates the production of VEGF in mast cells, IL-12 in DC and macrophages, as well as superoxide and chemokine synthesis in neutrophils [54]. It also induces the release of oxygen radicals through NK2R [55].

A preclinical study demonstrated that topical NK-1 receptor antagonist fosaprepitant significantly reduces established corneal neovascularization through dual anti-angiogenic and anti-inflammatory mechanisms. These findings position NK-1 receptor blockade as a promising therapeutic strategy for sight-threatening corneal neovascular disorders, particularly given its ability to modulate both vascular and immune components of the pathology [56].

As mentioned before, SP promotes angiogenesis by upregulating vascular endothelial growth factor (VEGF). In mouse models, rapamycin treatment reduces SP and VEGF levels. These findings suggest rapamycin could serve as a potential anti-angiogenic therapy for corneal neovascularization disorders [57].

## 5. Substance P and Interactions with Other Cytokines

As mentioned earlier, SP functions as a mediator in neuro-immune communication. Through its interaction with various chemokines and cytokines, it modulates a wide range of immunological processes. For example, the upregulation of IL-2 results in direct stimulation of immunoglobulin production. Furthermore, via NK1R, substance P increases the half-life of IL-8 transcripts in human corneal epithelial cells, leading to enhanced IL-8 synthesis. Similarly, primary human keratocytes expressing NK1R secrete higher level of chemotactic IL-8 protein upon SP stimulation. The upregulated expression of this pro-inflammatory and chemotactic cytokine significantly contributes to SP-enhanced keratocyte migration and can attract neutrophils [58]. Clinical studies associate elevated tear SP and NGF levels with dry eye symptoms [59], highlighting its diagnostic and therapeutic relevance.

Substance P is known to regulate both pro- and anti-inflammatory cytokines, maintaining the epithelial barrier against infections. It stimulates natural killer (NK) cells to produce IFN-gamma, which has a protective effect against bacterial infections. Downregulation of the mTOR pathway results in increased expression of pro-inflammatory cytokines IL-12p40 and IL-21, as well as decreased expression of IL-10 [60].

Another interesting fact is that substance P induces mast cell degranulation and subsequent secretion of various pro-angiogenic factors, such as vascular endothelial growth factor and TNF-alpha [43]. More information about its pro-angiogenic function can be found in another chapter of this article.

## 6. The Role of SP in Epithelial Homeostasis

As mentioned earlier, the cornea is highly innervated with sensory nerves that produce SP and cGRP neuropeptides. The neuronal cell bodies of these nerves are localized in the trigeminal ganglia (TG).

Electrical stimulation of the trigeminal ganglia has been shown to increase tear secretion, which depends on the release of SP from the sensory nerve endings [61]. In addition to trigeminal sensory neurons, corneal epithelial cells, stromal keratocytes, and immune cells also secrete SP at the ocular surface [10].

Several studies have explored the presence of SP and its metabolites in normal human tears and their role in maintaining corneal epithelial homeostasis [62].

The release of Substance P from corneal nerve endings mediates reflex tear production [61]. Moreover, studies on mice have shown that individuals genetically deficient in functional NK1R have reduced levels of basal tears in comparison to wild type B6 mice, as measured by the phenol red thread test, and develop clinical features associated with dry eye disease [63].

Another important aspect is the interaction between SP-NKR1, E-cadherin and zonula ocludens-1 tight junction proteins in corneal epithelial cells. The regulation of these proteins suggests a protective role of SP-NKR1 on preserving the corneal epithelial barrier [64]. By enhancing SP insulin-like growth factor-1 (IGF-1), fibronectin, and IL-6 to promote corneal epithelial migration, regulating the regeneration of the corneal epithelium and maintaining its integrity [65]. Additionally, SP modulates epithelial cell attachment to the extracellular matrix, promoting corneal wound healing.

In a diabetic mouse model, SP has been reported to promote corneal epithelial wound healing via NK1R receptor [66]. In an alkali-burn model in mice and rabbits, SP was shown to mobilize CD29^+^ stromal cells from the bone marrow into the circulation and subsequently to the injured tissue, accelerating the wound healing process [67,68].

Another interesting area of investigation is the potential synergistic effect of combination of substance P and insulin-like growth factor-1. Chikama et al. demonstrated complete recovery of neurotrophic keratitis in a patient with a combination of SP and insulin-like growth factor-1 (IGF-1) eye drops. They hypothesized that the combination of SP and IGF-1 stimulates corneal epithelial cell migration and the expression of integrin α5 and β1, which are essential for the attachment of epithelial cells to the extracellular matrix proteins [69]. In rabbits, the administration of eye drops containing both SP and IGF-1 promoted corneal epithelial wound healing. Furthermore, this synergistic effect of substance P and IGF-1 was found to be mediated through the NK-1 receptor for substance P on corneal epithelial cells [70]. SP/IGF-1 treatment was also reported to improve the barrier function and enhance epithelial wound healing in animal models of neurotrophic keratopathy [70,71].

Because substance P is easily degraded and inactivated by peptidases in the body [72], some researchers have attempted to identify the minimum essential amino-acid sequence of substance P to develop the clinical use of substance P for treating neurotrophic keratopathy. Their results demonstrated that Phe-Gly-Leu-Met-NH_2_, the sequence of four amino acids from the C-terminal of substance P, is the minimum sequence necessary to generate a synergistic effect with IGF-1 on corneal epithelial migration in vitro and epithelial wound closure in vivo [73].

Several studies have concluded that SP inhibits corneal epithelial cell apoptosis induced by hyperosmotic stress through Akt activation [74]. Furthermore, in mice deficient in NK1R, excessive exfoliation of apical corneal epithelial cells was observed, strongly suggesting that SP expression is critical for maintaining corneal epithelial integrity [63].

As mentioned earlier, substance P plays a critical role in preserving homeostasis of the ocular surface by influencing cytokine and chemokine production. It regulates both pro and anti-inflammatory processes, thus playing multi-faceted roles on the ocular surface physiology. In fact, future studies are needed to investigate and hopefully better understand the dichotomous nature of SP.

## 7. The Role of Substance P in Neurotrophic Keratitis

The most common causes of corneal anesthesia include viral infection (herpes simplex and herpes zoster keratoconjunctivitis), chemical burns, physical injuries, and corneal surgeries [75]. Intracranial space-occupying lesions, such as neuromas, meningiomas, and aneurysms, may also compress the trigeminal nerve or ganglion, leading to impaired corneal sensitivity [76]. Systemic diseases, such as diabetes, multiple sclerosis, and endemic leprosy can decrease sensory nerve function or damage sensory fibers, resulting in corneal anesthesia [77]. The corneal epithelium is the first target of the disease, showing dystrophic changes and defects with poor tendency to heal spontaneously. The progression of the disease may lead to corneal ulcers, melting, and perforation [77]. While the clinical diagnosis is relatively straightforward based on history and clinical findings, the management of this condition is one of the most challenging among all corneal diseases [58].

Studies have confirmed the healing effects of Substance P in diabetic corneal epitheliopathy [66]. In an alkali-burn model, the wound healing properties of SP were revealed due to the mobilization of bone marrow CD29^+^ stromal cells into the circulation and to the site of corneal injury [67]. Recently, use of both SP and IGF-1 in patients with neurotrophic keratopathy has shown promising outcomes [78].

The promotion of corneal epithelial cell migration initiated by the combination of SP and IGF-1 occurred in a dose-dependent manner, primarily mediated by NK1R, protein kinase C, and p38 MAPK activation pathway [79,80]. The efficacy of this combination was also demonstrated in rabbits following photorefractive keratectomy. In this study, topical use of substance P-IGF-1 combination significantly accelerated the rate of epithelial healing [81].

On the other hand, recent studies have revealed the role of SP in promoting fibrotic changes in the cornea. Increased production of collagen I, III, and V, lumican, α-Smooth Muscle Actin (SMA), fibronectin, and corneal fibroblasts contraction was observed, which is an undesirable effect in corneal wound healing [58].

In some studies, the potential role of nerve growth factor (NGF) in the treatment of neurotrophic keratitis has been evaluated [82]. NGF induces in vitro and in vivo recovery of sensory neurons and stimulates the production of Ach in the central nervous system and SP in the peripheral nervous system [83]. Additionally, NGF plays an important role in balancing sensory and sympathetic innervation by modulating their functions [83]. In rodents, corneal sensory innervation depends on NGF action, and in vitro, NGF promotes the proliferation and differentiation of rabbit corneal epithelial cells [84]. In other studies, topical administration of NGF in patients with neurotrophic ulcers restored corneal integrity and improved corneal sensitivity [85]. Recently, the REPARO trial showed that topical recombinant human nerve growth factor (rhNGF) eye drops restored corneal integrity in moderate-to-severe neurotrophic keratitis [86]. Furthermore, a clinical trial investigating recombinant human nerve growth factor (cenegermin) in moderate-to-severe dry eye disease demonstrated sustained improvement in patient-reported symptoms throughout the follow-up period [87].

In summary, damage to corneal sensory nerves leads to significant changes in the levels of neuromediators, resulting in impaired epithelial cell vitality, which clinically manifests as recurrent or persistent epithelial defects (Figure 1).

## 8. The Role of Substance P in Complications Due to Refractive Procedures

Refractive surgeries have a neurotrophic deprivation effect on the cornea due to the cutting of corneal sensory nerves, particularly after flap procedures [89].

Changes in neuromediators profiles and ocular surface parameters following laser refractive surgery vary depending on the surgical techniques and the severity of tissue damage.

Recent studies have investigated changes in tear neuromediators and corneal subbasal nerve plexus following small incision lenticule extraction (SMILE) as well as LASIK. In one study, corneal nerve fiber density (CNFD), length (CNFL), and corneal nerve branch density (CNBD) decreased significantly postoperatively, but tear levels of substance P and CGRP levels remained stable over 12 months [90]. Another study compared neuromediators profiles after SMILE and LASIK and found that SP level was significantly increased after LASIK but only moderately increased in the SMILE group [91]. The higher levels of tear substance P and NGF post-LASIK compared to SMILE may reflect differences in corneal innervation disruption between the two procedures. LASIK involves creating a corneal flap, which severs a greater number of subepithelial nerve fibers, triggering a stronger neuroinflammatory response and subsequent release of these neuropeptides. In contrast, SMILE’s flapless technique and smaller incision size likely preserve more nerve structures, leading to milder neurotrophic changes [92,93,94]. Importantly, a clinical cross-sectional study revealed tear SP levels positively correlate with severity of dry eye syndrome and negatively correlate with corneal sensitivity in patients after LASIK [95]. A comparison of corneal subbasal nerve morphology, corneal sensation, and tear film parameters after femtosecond lenticule extraction (FLEX) and small incision lenticule extraction (SMILE) showed that the less invasive SMILE technique better preserved central corneal nerves compared to FLEX. Moreover, corneal sensation was significantly reduced only in FLEX eyes. However, there were no differences between FLEX and SMILE when comparing tear film evaluation tests six months after surgery [91].

Future work should investigate whether these molecular differences correlate with long-term neural recovery or clinical outcomes. Some studies suggests that the impact on corneal nerves following refractive surgery is long-lasting. Although SMILE has better nerve preservation and regeneration than LASIK, neither procedure had recovered nerve status to normal levels even at 5.5 years post-surgery [92].

Future directions, including the use of neuromediators as potential biomarkers for ocular surface health following laser refractive surgery, are not fully understood and still require further investigation.

## 9. Interactions Between Dopamine, Substance P and Corneal Epithelium

Recently, the presence of monoamine receptors, which are found throughout the body, has been investigated in corneal epithelium. Antibodies to alpha-1, beta-1 and beta-2 adrenergic receptors, as well as to D1-like and 5HT-7 receptors, were found in the corneal epithelium. These receptors may play a role in regulating fluid transport or corneal homeostasis [96].

In another study, dopamine receptors in bovine corneal epithelium were investigated. It was found that the corneal epithelium and endothelium express a functional D1-like receptor positively coupled to adenylyl cyclase and cAMP production. However, the physiological role of this receptor remains speculative [97].

The release of Substance P (SP) is bidirectionally regulated by dopamine receptors, with D1 and D2 subtypes exerting opposing effects. D1 receptors enhance SP release through multiple mechanisms: increasing dendritic excitability and glutamatergic signaling in striatonigral medium spiny neurons (MSNs) of the direct pathway [98], promoting synaptic plasticity that potentiates sustained SP release [98] and facilitating reward processing and motor control [99]. Conversely, D2 receptor activation suppresses SP release by inhibiting glutamatergic transmission in striatopallidal MSNs of the indirect pathway [98].

In summary, dopamine D1 and D2 receptors work together in different ways to control Substance P release. How they interact depends on several factors, such as neuronal pathway, synaptic plasticity, and the presence of heteromers [100].

## 10. Discussion

The cornea, as the first refractive element of the eye, plays a critical role in vision and ocular health [1] Substance P (SP), a neuropeptide, has emerged as a significant player in corneal physiology, influencing processes such as pain perception, immune response, and epithelial homeostasis [12,101,102].

### 10.1. Corneal Innervation and Homeostasis

The cornea’s unique structure, characterized by its rich innervation, highlights the importance of sensory nerves in maintaining corneal integrity [1,22]. The relationship between corneal nerves and epithelial cells, as described, indicates that sensory innervation is crucial for delivering trophic factors that sustain epithelial health The loss of corneal sensory innervation, resulting in decreased epithelial vitality and increased susceptibility to damage, emphasizes the need for a deeper understanding of neurotrophic factors like SP in corneal homeostasis [12].

### 10.2. Pain Perception and Inflammatory Response

SP’s role in corneal pain perception is noteworthy, particularly in conditions like dry eye and infectious keratitis [7]. The findings suggest that elevated SP levels correlate with increased nociception, indicating its potential as a biomarker for pain assessment in corneal diseases [101]. The dual role of SP as both a pro-nociceptive and, in some contexts, an anti-nociceptive agent underscores the complexity of pain mechanisms in the cornea. This complexity necessitates further investigation into the crosstalk between SP and other neuropeptides, such as CGRP and the TRPV1 receptor, to better understand their combined effects on corneal sensory pathways [23].

### 10.3. Immune Modulation and Inflammation

The immunological role of SP, as highlighted in the paper, reveals its involvement in mediating inflammatory responses [16]. The interaction between SP and various immune cells indicates that it may serve as a bridge between the nervous and immune systems, influencing both innate and adaptive immunityThe upregulation of chemokines and cytokines by SP facilitates immune cell recruitment, suggesting potential therapeutic avenues for managing corneal inflammation, particularly in viral infections and allergic conditions.

### 10.4. Corneal Neovascularization and Wound Healing

The paper discusses the pro-angiogenic effects of SP, linking it to corneal neovascularization. This relationship is particularly relevant in pathological conditions where corneal transparency is compromised. Understanding the molecular pathways through which SP promotes angiogenesis could provide insights into potential therapeutic strategies to mitigate unwanted neovascularization while enhancing wound healing processes. The combination of SP and growth factors like IGF-1, as mentioned, presents a promising approach for treating neurotrophic keratitis and other corneal injuries.

The therapeutic potential of targeting SP and its receptors (NK1R, NK2R, NK3R) offers exciting prospects for ocular surface diseases [66,88]. However, the dichotomous nature of SP’s effects necessitates caution in therapeutic applications. Future research should focus on delineating the specific pathways activated by SP in various contexts, as well as exploring the potential of NK1R antagonists and other receptor modulators. Understanding the interplay between SP and other neuromediators, such as dopamine, could also unveil new therapeutic targets [98,99].

In conclusion, Substance P plays a multifaceted role in corneal physiology and pathology, influencing pain, immune response, and epithelial homeostasis. Continued exploration of SP’s functions and interactions within the corneal microenvironment will be essential in developing targeted therapies for a range of ocular surface diseases. The complexity of SP signaling pathways presents both challenges and opportunities for future research, creating opportunities for innovative approaches to enhance corneal health and restore integrity in pathological conditions.

## 11. Conclusions

Substance P is a tachykinin ubiquitous in the nervous and immune systems. It has therapeutic potential for various ocular surface diseases, as it is essential for maintaining ocular homeostasis. However, as mentioned earlier in this review, the dichotomous nature of SP remains a major challenge and is not fully understood. On one hand, it plays a crucial role in promoting corneal wound healing by enhancing keratocyte migration, while on the other hand, it amplifies inflammatory response at the ocular surface. Furthermore, there is a direct correlation between SP expression and severity of ocular surface inflammation, as clearly demonstrated by numerous studies. Initial evidence suggests that its healing properties are dose- and time-dependent. Further research is necessary to better understand the impact of these features of substance P on ocular surface diseases and to develop therapeutics that can effectively target them. TRPV1 activation triggers SP release from corneal nerves, amplifying inflammatory cascades. This neurogenic loop sustains ocular surface hypersensitivity in dry eye disease and post-surgical neuropathies. Therapeutic targeting of this axis (via NK1R antagonists/TRPV1 blockers) shows promise but requires further validation in corneal models. Recently, NK1R blockers—lanepitant and fosaprepitant—have been used in animal models to inhibit angiogenesis and inflammation. However, their therapeutic potential remains limited due to insufficient knowledge about factors influencing NK1R expression. Moreover, most studies focus on exploring substance P functions via NK1R. In fact, physiological and pathological responses generated by SP through its other receptors—NK2R and NK3R may have promising outcomes. Thus, more studies are essential and could shed light on the therapeutic applications of SP and other substances targeting its receptors.

## Figures and Tables

**Figure 1 biomolecules-15-00729-f001:**
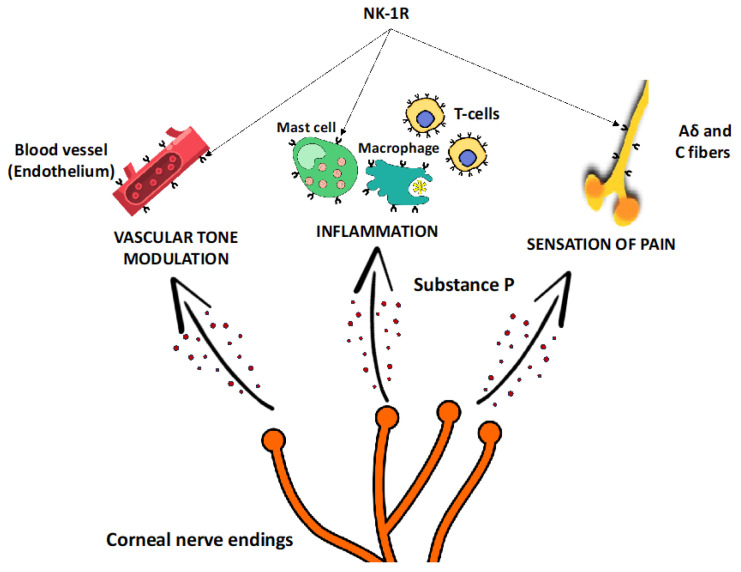
**Substance P interactions.** The influence of Substance P on vascular tone modulation, inflammation and pain perception [88].

## Data Availability

This is a review article, and no new datasets were generated or analyzed. All data discussed in this manuscript are available in the cited references.

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
