# Peer review of "The Role of Substance P in Corneal Homeostasis"

_biomolecules, 2025, doi:10.3390/biom15050729_

Round 1
Reviewer 1 Report
Comments and Suggestions for Authors
Summary
This review focuses on the role of substance P (SP) specifically in corneal homeostasis. Substance P is a small protein-like molecule of the tachykinin family and plays several key roles in the body, particularly in relation to pain perception and inflammation at the neuronal level. It is interesting to note that the pain receptor density in neuronal fibers of the cornea is the highest in the human body. The focus here is on the involvement of substance P (via NK1R) in corneal physiology (e.g. in corneal innervation, barrier function, immune regulation, wound healing) and pathophysiology in connection with various eye diseases with altered SP levels such as neurotrophic keratopathy, dry eye, corneal neovascularization, allergic conjunctivitis and also after post-refractive surgery (e.g. LASIK), which correlates with nerve damage. Even (benign) tumor diseases such as pterygium are discussed with regard to altered expression of SP and NK1R. Clinically, attempts are being made to develop or research therapies with NK1R antagonists. Also, combination therapies with IGF-1 could be promising for neurotrophic ulcers, for example. While many studies focus on the NK1R-mediated signaling pathways, there is still a need for research on NK2R/NK3R signaling pathways. Overall, closer investigation of SP mechanisms could significantly improve the development of therapies for corneal diseases (especially sensory neuropathies).
General Comments
The well-written manuscript has a generally good structure, with a detailed introduction on substance P and further chapters focusing on the role of SP in corneal pain perception, immune response, corneal neovascularization, interactions with cytokines and dopamine, epithelial homeostasis, neurotrophic keratitis, and even complications after refractory surgical procedures. This provides a good overview of the current state of research and also highlights areas still requiring further research. Since various corneal diseases are described here, it might be helpful to summarize the role and mechanism of SP in an overview table. Figure 1 fits well and could also be used as a graphical abstract. However, this manuscript does have minor deficiencies. For example, it lacks considerations of certain aspects of pain research, which includes corneal pain, since the cornea is known to have the highest density of pain receptors in the human body (PMID 12697417). Of particular note here is the TRPV1 pain receptor, also known as capsaicin receptor (PMID 28118665). Although two studies [17 and 23] are cited in connection with capsaicin and SP, the current state of research on the actual involvement of SP and TRPV1 in corneal innervation is not described in detail (compare PMID 29867077, PMID 16564032, PMID 28118665). Therefore, it would be beneficial if this aspect of (pain) research could be described in a (further) subchapter. The work of Belmonte et al. was groundbreaking here. Another deficiency is the omission of the role of SP at the non-neuronal level or in non-neuronal corneal epithelial cells, which are known to be unable to trigger action potentials but can instead trigger pro-inflammatory cytokines (PMID 17397832). For example, TRPV1 signaling is required for the upregulation of IL-6 and substance P and the healing of debrided corneal epithelium in mice (PMID 24781945). These and similar important findings in connection with pain receptors should definitely be embedded into the manuscript.
Specific comments
Abstract
Line 26: The abbreviations LASIK and SMILE in context with post-refractive surgery should either be written out here or omitted.
Line 23: The abbreviation IGF-1 should be written out here. It can then remain in line 28.
Lines 32-33: The statement that SP could revolutionize the treatment of corneal diseases (e.g. sensory neuropathy) could seem exaggerated (dichotomous nature of SP), especially since the interaction of other factors (such as the capsaicin receptor described above) could be relevant here. Suffice it to say, for example, that research into the multi-layered mechanisms of SP could well advance the development of therapies for corneal diseases.
- Introduction
Line 51-52: “Corneal epithelial cells secrete neurotrophic factors (e.g. nerve growth factor and ciliary neurotrophic factors) that affect corneal nerves.” It would be good to provide a source here.
- The role of SP in corneal pain perception
Line 96ff: In this section, at least the work on corneal innervation by Belmonte, Marfurt and others should be mentioned (PMID 31471469, PMID 20036654, PMID 12697417) also in connection with SP (PMID 16564032, 24781945, 29867077).
- The role of SP in regulating immune response
Line 145ff: “Several studies have demonstrated a surprising role of substance P and the neurokinin-1 receptor in the host response in viral infections.” It would be good to provide a source here.
Line 165-165: Subconjunctival injections of SP have also been shown to cause conjunctivitis and increase the permeability of conjunctival blood vessels.” Here too, a reference is missing.
- SP in corneal neovascularization
It would be interesting to add here what treatment concepts for this disease could look like. For example, the protein level of substance P (and VEGF) is significantly decreased in mice treated with rapamycin than the control mice (PMID 15671269). I recommend supplementing this section accordingly.
- SP and interactions with other cytokines
Line 191: “…higher level of chemotactic IL-8 protein upon SP stimulation.” I recommend considering clinical aspects of dry eye in this section. For example, cytokines, SP and NGF have been associated with pain and dry eye (see tear concentrations of various cytokines and SP) (PMID 35358536).
- The role of SP in epithelial homeostasis
Line 235ff: The references are missing in the first sentence. How do you explain the controversial findings after SP treatment?
Line 240-241, 285-286: Are there other combined therapeutic approaches besides IGF-1?
- Complications due to refractive procedures
Line 333-334: There is no reference in the first sentence of this paragraph. How could one explain the increased SP and NGF tear level in the LASIK group?
- Conclusions
The conclusions should be supplemented accordingly with regard to the pain receptors mentioned in the general comments (see above: …current state of research on the actual involvement of SP and TRPV1 in corneal innervation). This also applies to chapter 2 (from line 96).
Minor
Substance P is already explained or spelled out in line 61. After that in lines 74, 85, 88, 96, 97, 100, 109, 111, 112, 122, 127, 129, 134, 141, 145, 166, 168, 176, 182, 188, 193, 198, 203, 218, 239, 246, 247, 251, 253, 255, 263, 282, 292, 333, 345, 356, 365, 370, it can be abbreviated, as has already been done in between.
References 53 and 54 are duplicates. Please check the reference list for duplicate entries.
Line 318: LASIK should be written out in full. After that, the abbreviation is sufficient.
Any papers recommended in the report are for reference only. They are not mandatory. You may cite and reference other papers related to this topic.
Reviewer 2 Report
Comments and Suggestions for Authors
This review article entitled "The role of Substance P in corneal homeostasis " has made a comprehensive literature review of substance P in corneal disorders and its dichotomous effects in both promoting tissue repair and exacerbating inflammation. However, several concerns are raised here:
- Many descriptions just cite the different experiment results without critical judging for readers.
- Several similar review articles can be found in the literature. Among them, this review manuscript and conclusion did not add new information than the paper cited by authors: Singh RB et al. Modulating the tachykinin: Role of substance P and neurokinin receptor expression in ocular surface disorders. Ocul Surf. 2022 Jul;25:142-153. doi: 10.1016/j.jtos.2022.06.007.
- Authors should make more descriptions of recent clinical experiments of substance p instead of citing a lot of old papers.
- Line 345 9. Interactions between dopamine, substance P and corneal epithelium. In this section, the description did not mention the relationship between the monoamine receptors and substance P
Reviewer 3 Report
Comments and Suggestions for Authors
This review focuses on the role of substance P in corneal homeostasis, exploring its mechanisms and clinical significance in various aspects such as corneal pain perception and immune regulation, but there is still room for optimization. Here are four review comments:
- Article figures and tables: As a review, there are few figures;
- Article structure: Lack of "discussion" section;
- Subtitle relevance: The expression of "7" and "8" is inconsistent with other subheadings;
- In depth exploration of mechanisms: Although it has been elucidated that substance P exerts its effects through multiple pathways, the upstream and downstream regulatory mechanisms of some key signaling pathways have not been thoroughly explored.
- Clinical relevance enhancement: It mentions the clinical research of substance P in various corneal diseases, but there is insufficient analysis of its feasibility and limitations in clinical application. More clinical case data are needed to evaluate the efficacy, safety, and potential risks of substance P as a therapeutic target in different corneal diseases.
- Literature citation: There are a large number of cited literature, but some of them are from a long time ago. Please appropriately increase the high-quality research achievements in recent years to reflect the latest research progress, making the content of the paper more timely and cutting-edge.
No major issues were found in the English language, please check for minor grammar problems.
Round 2
Reviewer 2 Report
Comments and Suggestions for Authors
Though authors have made an extensive revision, there are still some concerns:
- A good review article needs critical judgment to help readers. For example, Line 351-358. Tear substance P and nerve growth factor levels were significantly higher in the LASIK group than in post-SMILE eyes. Authors should find a better explanation in the literature.
- Line 210-214 Substance P also induces the production of another cytokine, IL-1. This results in bone marrow stimulation and the promotion of hematopoiesis. Several studies have shown that SP upregulates the chemotactic cytokines such as CCL4, CXCL2, MCP-1, and CCL5, which recruit monocytes and lymphocytes to sites of inflammation (Guo et al., 2002b). Is this action related to the corneal inflammation? If not, authors should revise it.
- Reference Guo et al., 2002b is similar to Guo et al., 2002a
- Please add some clinical trials to describe some drugs related to SP used in corneal disorders, and show their results to add value to this review paper
